# Efficacy of LD Bio *Aspergillus* ICT Lateral Flow Assay for Serodiagnosis of Chronic Pulmonary Aspergillosis

**DOI:** 10.3390/jof8040400

**Published:** 2022-04-14

**Authors:** Animesh Ray, Mohit Chowdhury, Janya Sachdev, Prayas Sethi, Ved Prakash Meena, Gagandeep Singh, Immaculata Xess, Surabhi Vyas, Maroof Ahmad Khan, Sanjeev Sinha, David W. Denning, Naveet Wig, Sushil Kumar Kabra

**Affiliations:** 1Department of Medicine, AIIMS, New Delhi 110029, India; chowdharymohit7439@gmail.com (M.C.); prayassethi82@gmail.com (P.S.); vpmahar05@gmail.com (V.P.M.); dr.sanjeevsinha@gmail.com (S.S.); naveetwig@gmail.com (N.W.); 2Department of Microbiology, AIIMS, New Delhi 110029, India; drjanyasachdev@gmail.com (J.S.); drgagandeep@gmail.com (G.S.); immaxess@gmail.com (I.X.); 3Department of Radiodiagnosis, AIIMS, New Delhi 110029, India; surabhi_vyas@yahoo.com; 4Department of Biostatistics, AIIMS, New Delhi 110029, India; khanmaroofahmad@gmail.com; 5Manchester Fungal Infection Group, Manchester Academic Health Science Centre, University of Manchester, Manchester M13 9NT, UK; david.denning@manchester.ac.uk; 6Department of Pediatrics, AIIMS, New Delhi 110029, India; skkabra@hotmail.com

**Keywords:** chronic pulmonary aspergillosis, aspergillosis, LDBio LFA, IgG, *Aspergillus*, post-tuberculosis, fungus, lateral flow assay, serodiagnosis, CPA

## Abstract

Background: The diagnosis of CPA relies on the detection of the IgG *Aspergillus* antibody, which is not freely available, especially in resource-poor settings. Point-of-care tests like LDBio *Aspergillus* ICT lateral flow assay, evaluated in only a few studies, have shown promising results for the diagnosis of CPA. However, no study has compared the diagnostic performances of LDBio LFA in setting of tuberculosis endemic countries and have compared it with that of IgG *Aspergillus*. Objectives: This study aimed to evaluate the diagnostic performances of LDBio LFA in CPA and compare it with existing the diagnostic algorithm utilising ImmunoCAP IgG *Aspergillus*. Methods: Serial patients presenting with respiratory symptoms (cough, haemoptysis, fever, etc.) for >4 weeks were screened for eligibility. Relevant investigations, including direct microscopy and culture of respiratory secretions, IgG *Aspergillus*, chest imaging, etc., were done according to existing algorithm. Serums of all patients were tested by LDBio LFA and IgG *Aspergillus* (ImmunoCAP Asp IgG) and their diagnostic performances were compared. Results: A total of 174 patients were included in the study with ~66.7% patients having past history of tuberculosis. A diagnosis of CPA was made in 74 (42.5%) of patients. The estimated sensitivity and specificity of LDBio LFA was 67.6% (95% CI: 55.7–78%) and 81% (95% CI: 71.9–88.2%), respectively, which increased to 73.3% (95% CI: 60.3–83.9%) and 83.9% (95% CI: 71.7–92.4%), respectively, in patients with a past history of tuberculosis. The sensitivity and specificity of IgG *Aspergillus* was 82.4% (95% CI: 71.8–90.3%) and 82% (95% CI: 73.1–89%); 86.7% (95% CI: 75.4–94.1%) and 80.4% (95% CI: 67.6–89.8%), in the whole group and those with past history of tuberculosis, respectively. Conclusions: LDBio LFA is a point-of-care test with reasonable sensitivity and specificity. However, further tests may have to be done to rule-in or rule-out the diagnosis of CPA in the appropriate setting.

## 1. Introduction

The relationship between the fungus and the host typically determines the manifestation of *Aspergillus* lung disease ranging from acute and subacute invasive to chronic pulmonary aspergillosis [1,2]. Chronic pulmonary aspergillosis (CPA) is a collection of serious illnesses characterised by persistent cough, dyspnoea, haemoptysis, fatigue, and weight loss [3]. Chronic cavitary pulmonary aspergillosis (CCPA) is the most frequent manifestation of CPA, which can progress to chronic fibrosing pulmonary aspergillosis if left untreated. Single aspergilloma and *Aspergillus* nodule are some of the less prevalent manifestations of CPA [4]. Pre-disposing factors for CPA include underlying pulmonary illnesses like mycobacteriosis or chronic obstructive pulmonary disease (COPD), as well as prevalent immunosuppressive disorders such as diabetes [3].

Patients with CPA have significant morbidity, affecting roughly 3 million individuals globally. The overall 5-year mortality rate ranges up to 80%, resulting in an estimated 450,000 annual fatalities [5]. As a result of India’s high TB disease load, post-tuberculosis sequelae are common. CPA has an annual incidence of 27,000 to 170,000 cases, with a 5-year prevalence of 24 per 100,000 [6]. In India, the prevalence may be very high in post-tuberculosis sequelae patients (~57%), and recurrence of tuberculosis has been reported to be an important independent risk factor for development of CPA [7]. No single clinical or radiological manifestation or laboratory result is adequate for a conclusive diagnosis of CPA; rather, an amalgamation of clinical, radiographic, and microbiological findings is used since the presentation may be non-specific and might be difficult to differentiate from pulmonary tuberculosis [8,9,10,11]. The microbiological evidence for diagnosis is considered with direct confirmation of *Aspergillus* infection (microscopy or culture from BAL fluid/biopsy) or an immune response to *Aspergillus* spp. [4].

In the frequent absence of positive cultures, serologic assays are essential for the diagnosis of CPA [12]. In the earliest assay formats, antibodies against *Aspergillus fumigatus* were identified by detection of precipitins with high specificity utilising the double immunodiffusion test (DID) or counter immune-electrophoresis (CIE) technique. These methods, however, had a long turnaround time, required significant labour with a large inoculum of fungal extract and patient serum extracts, and the results were only semi-quantitative [12]. Other commercially accessible serological tests were subsequently launched and may be employed for diagnosis of CPA, such as enzyme immunoassay (EIA), enzyme-linked immunosorbent assay (ELISA), and indirect hemagglutination (IHA); however, performance levels vary amongst tests, and redefining cut-off values for different populations and diagnoses may be required to maximise performance [8]. Out of these tests, indirect hemagglutination is clearly inferior to other methods [13]. Moreover, the performance of these tests often requires sophisticated equipments, steady power supply, and technical expertise, besides considerable costs.

In recent years, the lateral flow assay (LFA) has been employed to simplify *Aspergillus* IgG detection with quick turn-around time and little laboratory equipment. The only commercially available LFA for detecting *Aspergillus* IgG is LDBio LFA *Aspergillus* immunochromatographic technology (hereafter referred to as LDBio LFA) [14]. A point-of-care test has been felt to be essential in simplifying the diagnosis and management of CPA, especially in a resource-limited setting. When compared to ImmunoCAP (Thermo Fisher, Waltham, MA, USA) for levels of *Aspergillus fumigatus* specific IgG (hereafter referred to as ImmunoCAP Asp IgG), the LDBio LFA has exhibited good sensitivity and specificity for detection of CPA in studies done in centres in France and United Kingdom [8,15]. A study had also reported sensitivity and specificity of 80% and 70%, respectively, for LDBio LFA from Indonesia, which, however, did not use an alternative IgG *Aspergillus* in the diagnostic algorithm. Thus, the present study was conducted out of the necessity of evaluating the diagnostic performances of this point-of-care test in a tuberculosis endemic country with a significant burden of CPA and to compare it with the existing diagnostic criteria including ImmunoCAP Asp IgG.

## 2. Materials and Methods

Between February 2020 and December 2021, consecutive patients presenting to the chest clinic of a tertiary care unit in North India with respiratory symptoms (cough, haemoptysis, fever, shortness of breath, chest pain, etc.) of more than four weeks duration were enrolled in the study. Patients with an apparent non-CPA diagnosis, such as lung malignancy or who refused consent for serological tests were excluded from the study. Ethical permission was taken from Institute Ethics Committee for the conduct of this study (IEC-52/08.01.2021, RP-10/2021). A written consent form was obtained from all participants before enrolment into the study as per institutional protocol.

After enrolment, the case details were recorded in a standardised case record form (CRF) by a trained professional. Relevant investigations including blood tests, chest imaging, sputum examination, and bronchoalveolar lavage were done as per the discretion of the treating physicians. Serum samples collected from all the individuals were evaluated by LDBio LFA assays and ImmunoCAP Asp IgG assay. The diagnosis of CPA was made individually by two researchers (AR and MC), and then corroborated as per the European Respiratory Society/European Society of Clinical Microbiology and Infectious Diseases (ERS/ESCMID) criteria [4]. The diagnosis relied on appropriate clinical, radiological, and microbiological parameters. Clinical parameters includes the presence of at least one of these symptoms, such as haemoptysis, cough, exhaustion, chest discomfort, and/or dyspnoea for more than three months. Radiographic findings consistent with CPA (at least one of cavitation and/or fungal ball confirmed by CT scan) should be present [16]. Microbiological evidence included a positive serological result using the ImmunoCAP Asp IgG assay to measure *Aspergillus*-specific IgG levels (>27 milligrams of antibodies/liter considered to be the cut-off for positive result), histopathological evidence of CPA following lung biopsy or resection, positive result in galactomannan assay performed on serum or BAL samples using the Platelia Aspergillus galactomannan ELISA (Bio-Rad Laboratories) interpreted according to cut-offs provided in the 2019 EORTC/MSGERC guidelines (galactomannan index >1 was considered positive for both serum and BAL), and respiratory samples showing hyaline septate hyphae morphologically suggestive of *Aspergillus* spp. in direct microscopy, or growth of *Aspergillus* spp. in culture [4]. Radiological criteria for diagnosis were adapted from those used commonly in a resource-constrained setting [8]. The *Aspergillus* ICT IgG IgM lateral flow assay (LDBio LFA, Diagnostics, Lyon, France) in a cassette format was used to test each sample, and all tests were performed and interpreted according to the manufacturer’s instructions. The required number of cassettes were removed from storage at 4 °C, brought to room temperature, and labelled. A measure of 15 µL of serum was added to the sample well of each cassette using a calibrated micropipette and sterile disposable tips, followed by four drops of eluent dispensed directly from the dropper. The cassettes were then allowed to stand for 20 min, and the test results were read between 20 and 30 min after adding the eluent to the last cassette. Accounting for pre-test centrifugation of blood samples to separate serum and proper pre- and post-test documentation, the turnaround time of the test (from receiving a blood sample in the laboratory to availability of the report online) was therefore between 30 and 45 min. The presence of a well-defined black line at the “Test (T)” and “Control (C)” markers was considered as a positive result. The presence of a thin, diffuse grey line at the “T” marker indicated a “weakly positive” result. An alternative analysis was done assuming cut-off of 40 mgA/L for ImmunoCAP Asp IgG, as is suggested by the manufacturer as well as validated in studies [8,17,18].

## 3. Statistical Analysis

The sample size was estimated considering a sensitivity of 85% and specificity of 84% (average of the diagnostic accuracies reported by Hunter et al. [8] in healthy population and Rozaliyani et al. [19] in tuberculosis-treated patients) and taking a precision of 8.5% with 95% confidence level. The sample size estimated was at least 68 cases and 72 non-CPA cases.

Categorical and continuous variables were reported in frequencies/percentages and mean with standard deviation/median with minimum, maximum depending on the nature of the data, respectively. Fisher’s exact/chi-squared tests were used to establish association for categorical variables, while Student’s *t*-test or Wilcoxon rank sum test was used for continuous variables as appropriate. Those with *p*-value < 0.05 were considered to be significant.

## 4. Results

A total of 218 patients were screened and 174 patients were included in this study (Figure 1). The baseline characteristics of the study population are detailed in Table 1, Table 2 and Table 3. One hundred and eight (62.1%) enrolees were male with a mean age of 40.7 (±13.9 years). The vast majority of patients (98.3%) were HIV seronegative. In total, 116 (66.67%) patients had a prior history of documented pulmonary tuberculosis and 46 (26.4%) had used inhalational devices in the last month.

In this study population, 74 patients (42.5%) fulfilled the diagnosis of CPA, while a diagnosis of allergic bronchopulmonary aspergillosis was made in 13 (7.5%) patients. Out of the 74 patients with CPA, 44 (59.5%) had chronic cavitary pulmonary aspergillosis, 19 (25.7%) had chronic fibrosing pulmonary aspergillosis, and 10 (13.5%) had *Aspergillus* nodules. The majority of the patients (>90%) were diagnosed by positive ImmunoCAP Asp IgG assay as the sole microbiological criteria along with corroborative clinic-radiological features. Pulmonary tuberculosis was diagnosed in 16 (9.2%) patients. Cough was the most common symptom, reported in 106 (60.9%) patients. On chest imaging, cavity was the most common reported finding, being present in 74 patients (42.5%). ImmunoCAP Asp IgG was elevated in 61 (82.4%) of the group diagnosed with CPA. Out of 131 patients who produced sputum or underwent bronchoalveolar lavage (BAL), the fungal culture showed growth of *Aspergillus* spp. in nine patients (6.9%). BAL galactomannan was positive in 17 (23%) patients of those with CPA.

The sensitivity and specificity of LDBio LFA for diagnosis of CPA (as compared to the ERS/ESCMID criteria) in our study subjects presenting with respiratory symptoms for at least four weeks were 67.6% (95% CI: 55.7–78%) and 81% (95% CI: 71.9–88.2%), respectively, with a diagnostic accuracy of 75.3%. In the population with a past history of tuberculosis, the sensitivity and specificity were 73.3% (95% CI: 60.3–83.9%) and 83.9% (95% CI: 71.7–92.4%), respectively, and the estimated diagnostic accuracy was 78.5%. In those with past history of tuberculosis and with symptoms >3 months, the sensitivity and specificity were 74.1% (95% CI: 60.3–85%) and 85% (95% CI: 70.2–94.3%), respectively, with the diagnostic accuracy being 78.7%.

In contrast, the sensitivity and specificity for ImmunoCAP Asp IgG for diagnosis of CPA were 82.4% (95% CI: 71.8–90.3%) and 82% (95% CI: 73.1–89%), respectively. The diagnostic performances of both LDBio LFA and ImmunoCAP Asp IgG are represented in Table 4. The kappa for agreement between LDBio LFA and *Aspergillus fumigatus* was 0.53.

According to the alternative analysis using the cut-off of Aspergillus-specific IgG assay at 40 mgA/L, the diagnostic performances of LDBio LFA were altered (in most cases the diagnostic accuracies marginally fell), as shown in Table 5. The chest radiograph and CT thorax of a CPA patient who was negative for both ImmunoCAP Asp IgG & LDBio LFA is depicted in Figure 2 Pictorial representation of positive & negative results of the LBDio assay are depicted in Figure 3.

## 5. Discussion

The benefits of LDBio LFA under evaluation reported in existing literature include minimal requirement of resources, time, and machinery—all of which are important in diagnosing CPA in the resource-constrained settings where CPA is predominantly found. In the present study, in a population presenting with symptoms predominantly suggestive of persistent respiratory symptoms (>four weeks), the assay had a sensitivity, specificity, and diagnostic accuracy of 67.6%, 81%, and 75.3%, respectively. In the population who had a past history of pulmonary tuberculosis, the sensitivity, specificity, and diagnostic accuracy of the assay increased to 73.3%, 83.9%, and 78.5%, respectively. The ImmunoCAP Asp IgG assay for detection of IgG *Aspergillus fumigatus* in the same population had a better sensitivity of ~82.4% and diagnostic accuracy of 82.2%, though with similar specificity (82%), using a cut-off of >27 mA/mL. This performance is not as good as what has been published previously at the same cut-off from a previous study [19]. The possible reason may be the inclusion of serum galactomannan (EIA > 0.5) as microbiological criteria for diagnosis of CPA in the previous study. A significant proportion of controls in our study were also positive for serum galactomannan, indicating a plausibility of false positivity owing to previous antibiotics or food habits [20,21]. There was moderate degree of agreement [22] between LDBio LFA and the ImmunoCAP Asp IgG assay (kappa = 0.53).

Diagnosis of CPA is dependent on the use of serological tests, especially IgG against *Aspergillus* spp. (e.g., ImmunoCAP Asp IgG), in addition to radiological features [4]. Since the antibody estimation is costly, access to these tests is often restricted in resource constrained settings [23]. On the other hand, conventional fungal cultures typically have a poor positivity rate (ranging from 10–40%) [24], which might increase markedly by high volume culture (~54% in mixed cases of pulmonary aspergillosis) [25]. Further, different features on chest radiographs and CT scans have variable sensitivity and specificity, which may be as low as ~28% [26]. For example, a normal chest radiograph had an excellent performance in ruling out CPA, and such patients may not need testing for *Aspergillus* IgG [26]. Utility of galactomannan antigen in serum or BAL have been evaluated in numerous studies, which have yielded different cut-offs, making it difficult to introduce uniform criteria for diagnosis of CPA [27,28,29]. Access to point-of-care tests like LDBio LFA is important in identifying the significant load of CPA patients in tuberculosis-endemic countries, which are usually “economically developing” and resource constrained. Our study shows that this assay can be used as a screening test to detect ~70% of CPA patients. However, the test may be false-negative in ~30% of CPA patients, implying that other ancillary tests have to be performed before ruling out the diagnosis of CPA in those patients with high index of suspicion. Our study also showed that the LDBio LFA and ImmunoCAP Asp IgG assay had a similar specificity of ~82%, suggesting that the former may be a reliable “rule-in” test. In light of the present study, it seems that LDBio LFA, though demonstrating lower sensitivity and comparable specificity with ImmunoCAP Asp IgG, may have a significant role in identifying patients with CPA. It can be used a screening test in patients presenting with persistent respiratory symptoms in whom CPA is a probable diagnosis. Due to the low sensitivity of LDBio LFA, those who are negative need to be followed up with other tests like ImmunoCAP Asp IgG. Those who are positive may be treated as cases of CPA, owing to the similar specificity of ImmunoCAP Asp IgG assay, provided the clinical and radiological features are compatible with the diagnosis of CPA. An algorithm depicting the possible role of LDBio LFA is depicted in Figure 4.

The effectiveness of the LDBio LFA in diagnosing CPA has been demonstrated in three previous studies—one each in France, the United Kingdom (UK), and Indonesia—and its use has also been reported in a case study from Uganda [30]. The reported sensitivity and specificity in the three studies along with that of our study is shown in Table 6. The differences in the sensitivities/specificities of the three studies can be possibly explained by the differences in the recruited population. While the Indonesian study included patients after completion of tuberculosis therapy, the UK study included sera of known CPA patients and used the “matched” sera of healthy controls. In our study, we recruited patients from outpatients and inpatient settings with predominantly chronic respiratory symptoms as the primary presenting complaint. Different diagnostic criteria were used in the various studies. Approximately 66.7% of our patients had past history of pulmonary tuberculosis. Our study population likely represented the real-life scenario wherein CPA suspects often present without a past history of respiratory illnesses and often are misdiagnosed as “smear-negative” tuberculosis [31].

Our study had the following limitations. Only three of our patients (<2%) were afflicted by HIV. The absolute number of patients with growth of *Aspergillus* spp. in their respiratory samples was also low (~5.2%) in our study population. However, in our study, all the diagnoses were confirmed using the existing guidelines for diagnosis of CPA and ImmunoCAP Asp IgG, which is widely regarded as a high-quality quantitative diagnostic test, as it was performed in all patients.

## 6. Conclusions

LDBio LFA has reasonable sensitivity and specificity for diagnosis of CPA and can be used in resource-poor settings due to its simplicity of process, minimal requirement for equipment and infrastructure, quick turn-around time, and low cost.

## Figures and Tables

**Figure 1 jof-08-00400-f001:**
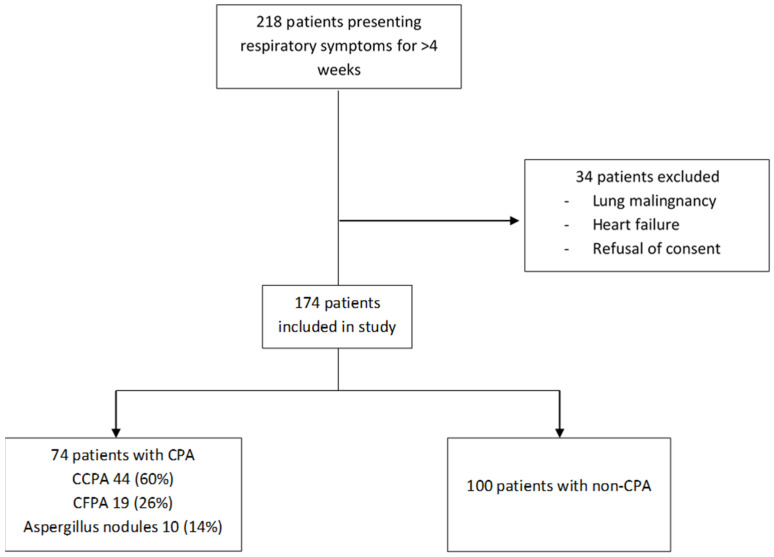
Enrolment in the study.

**Figure 2 jof-08-00400-f002:**
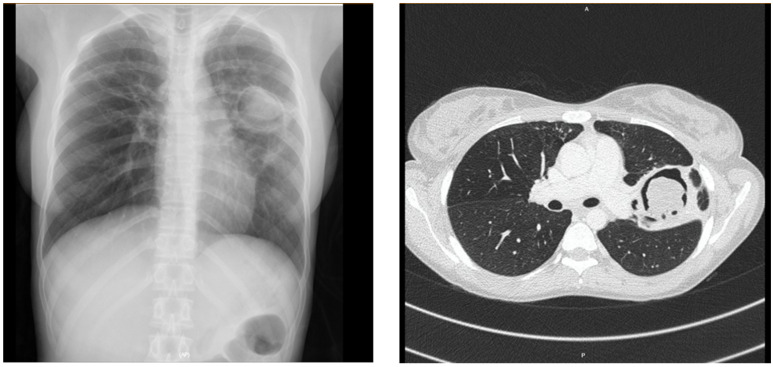
A 23 years old female patient with chest radiograph and CT thorax showing fungal ball in left upper lobe cavity. Her LDBio LFA and IgG were both negative.

**Figure 3 jof-08-00400-f003:**
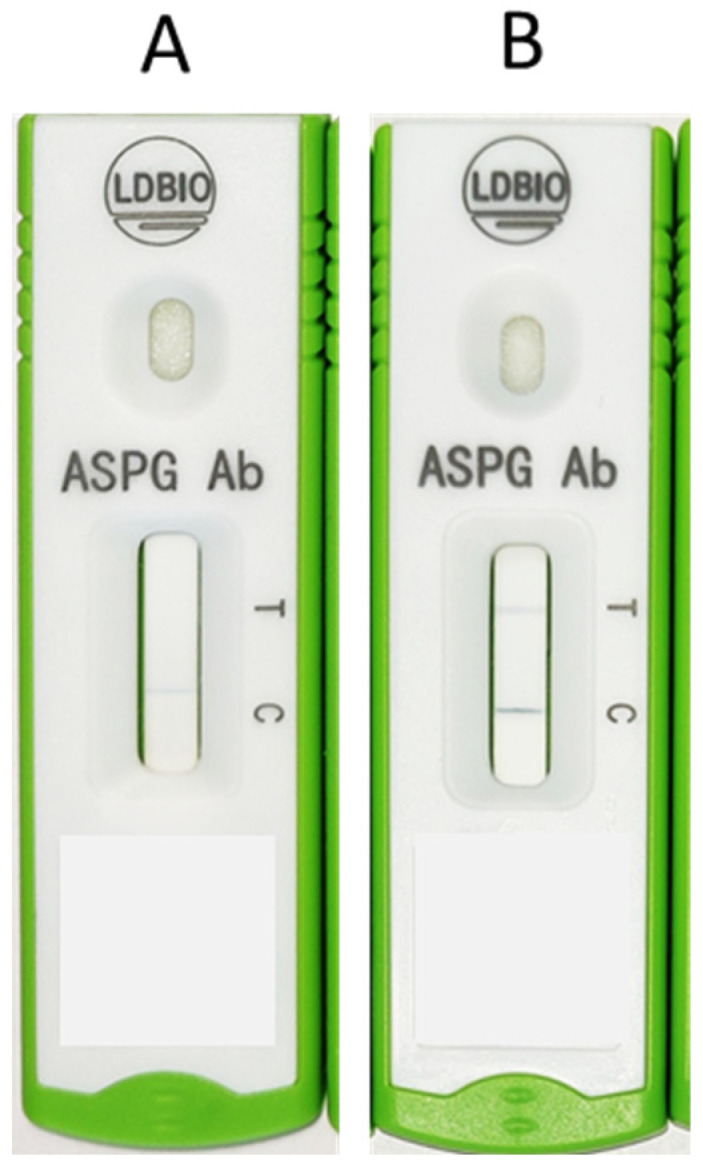
Pictorial representation of LDBio assay results: (**A**) negative, (**B**) positive.

**Figure 4 jof-08-00400-f004:**
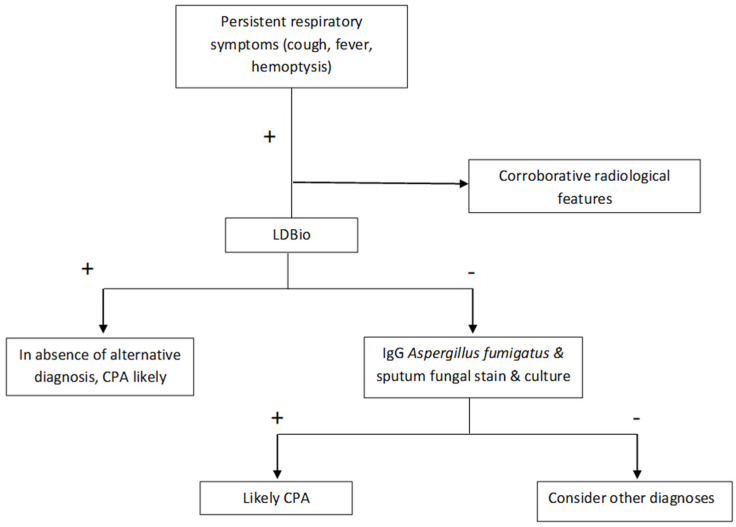
Proposed algorithm for utilising LDBio LFA in the diagnostic algorithm.

**Table 1 jof-08-00400-t001:** Comparison of baseline demographics.

Variables		Non-CPA (*n* = 100)	CPA (*n* = 74)	*p*-Value
Age (±SD)		41.4 (±14.57)	39.68 (±13.05)	0.42
Sex	Male	66 (66.0%)	42 (56.8%)	0.21
	Female	34 (34.0%)	32 (43.2%)	
Past History				
Tuberculosis		56 (56.0%)	60 (81.1%)	0.001
HIV		2 (2.0%)	1 (1.35%)	>0.99
ABPA		12 (12.0%)	1 (1.35%)	0.008
MDI use		28 (28%)	18 (24.3%)	0.67
Symptoms & Mycobacterial Workup				
Fever		10 (10%)	13 (17.6%)	0.16
Cough		53 (53%)	53 (71.6%)	0.02
Weight loss		25 (25%)	25 (33.8%)	0.24
Breathlessness		46 (46%)	31 (41.9%)	0.51
Haemoptysis		17(17%)	34 (46%)	<0.001
Fatigue		0 (0%)	2 (100%)	0.33
Any symptom		84 (84.0%)	68 (91.9%)	0.12
No. (%) with Post-TB sequelae		29 (29%)	52 (70.3%)	
No. (%) with pulmonary/disseminated TB		21 (21%)	9 (12.2%)	
No. (%) with ABPA		11 (11%)	1 (1.4%)	
No. (%) with obstructive airway disease		13 (13%)	2 (2.7%)	
No. (%) with sarcoidosis		5 (5%)	0 (0%)	
No. (%) with ILD		1 (1%)	1 (1.4%)	
No. (%) with others - Lung malignancy/metastasis - Lung mass under evaluation - Pulmonary/disseminated cryptococcosis - Pulmonary/disseminated mucormycosis - Post COVID sequelae - Gujjar’s lung - Chronic hypersensitivity pneumonitis - PUO - Bronchiectasis under evaluation - Diabetes mellitus - CML - Unclassified		4 (4%) 4 (4%) 1 (1%) 1 (1%) 2 (2%) 1 (1%) 1 (1%) 1 (1%) 5 (5%) 2 (2%) 0 (0%) 11 (11%)	0 (0%) 1 (1.4%) 0 (0%) 0 (0%) 0 (0%) 0 (0%) 0 (0%) 0 (0%) 0 (0%) 2 (2.7%) 1 (1.4%) 8 (10.8%)	

CPA: chronic pulmonary aspergillosis, ABPA: Allergic bronchopulmonary aspergillosis, MDI: Metered dose inhaler (Short & long acting bronchodilators/inhaled corticosteroids), ILD: Interstitial lung disease, PUO: Pyrexia of unknown origin, CML: Chronic myeloid leukaemia.

**Table 2 jof-08-00400-t002:** Comparison of imaging findings (Chest radiograph & CT chest).

Features	Non-CPA (*n* = 100)	CPA (*n* = 74)	*p*-Value
Consolidation	33 (33%)	35 (47.3%)	0.33
Cavity	22 (22%)	61 (82.4%)	<0.001
Nodules	40 (40%)	42 (56.8%)	0.14
Ground glass opacities	22 (22%)	26 (35.1%)	0.21
Pleural effusion	16 (16%)	5 (6.8%)	0.06
Pleural thickening	10 (10%)	24 (32.4%)	<0.001
Bronchiectasis	36 (36%)	46 (62.2%)	0.001

**Table 3 jof-08-00400-t003:** Comparison of laboratory parameters.

Variables	Non-CPA (*n* = 100)	CPA (*n* = 74)	*p*-Value
LDBio LFA positive *Weakly positive*	19 (19%) 2 (2%)	50 (67.6%) 1 (1.4%)	<0.001 0.6
Specific IgE (IQR)	0.11 (0.02–0.37)	0.34 (0.04–1.5)	0.2
Total IgE (IQR)	328 (66.9–1465)	406 (119–1035)	0.8
Specific IgG (IQR)	11.5 (6.40–20.1)	53.85 (30–91)	<0.001
AEC (IQR)	162.07 (49.6–341.66)	162.32 (15.19–371.46)	0.93
Specific IgE (≥0.1 KVA/L)	16 (16/28, 57.1%)	30 (30/46, 65.2%)	0.49
Total IgE (≥500 KVA/L)	14 (14/31, 45.2%)	20 (20/47, 42.6%)	0.82
Specific IgG (≥27 MgA/L)	18 (18.0%)	61 (82.4%)	<0.001
AEC (≥500 cells/mm3)	13 (13/60, 21.7%)	10 (10/50, 20%)	0.83
Positive direct KOH	4 (4/55, 7.3%)	6 (6/53, 11.3%)	0.52
Positive fungal culture *Aspergillus fumigatus* Aspergillus flavus Aspergillus niger Aspergillus spp.	2 (2/70, 2.9%) - 2 * 1 * -	7 (7/61, 11.5%) 2 3 1 1	0.08
Serum galactomannan (≥1.0)	10 (10/28, 35.7%)	16 (16/39, 41%)	0.66
BAL galactomannan (≥1.0)	7 (7/30, 23.3%)	17 (17/28, 60.7%)	0.004
ZN AFB + ve	2 (2/48, 4.2%)	0 (0/41, 0%)	0.49
MGIT + ve	5 (5/30, 16.7%)	0 (0/25, 0%)	0.06
GeneXpert	9 (9/55, 16.4%)	4 (4/54, 7.4%)	0.24
Any TB investigation + ve	12 (12/100, 12.0%)	4 (4/74, 5.4%)	0.19

* One patient showed mixed growth of *Aspergillus niger* and *Aspergillus flavus.* IQR: Interquartile range. AEC: absolute eosinophil count. ZN AFB: ZIEHL-NEELSEN acid-fast bacillus. MGIT: Mycobacteria Growth Indicator Tube.

**Table 4 jof-08-00400-t004:** Sensitivity & specificity of LDBio LFA kit & ImmunoCAP Asp IgG (cut off >27 mgA/L).

Population/Test	No of Observations			Sensitivity	Specificity	Diagnostic Accuracy
Symptoms > 4 weeks						
LDBio LFA		CPA	Non-CPA	67.6% (55.7–78%)	81% (71.9–88.2%)	75.3% (68.19–81.50%)
	LDBio neg	24	81			
	LDBio pos	50	19			
ImmunoCAP Asp IgG		CPA	Non-CPA	82.4% (71.8–90.3%)	82% (73.1–89%)	82.2% (75.68–87.56%)
	IgG neg	13	82			
	IgG pos	61	18			
Symptoms > 4 weeks with past history of PTB						
LDBio LFA		CPA	Non-CPA	73.3% (60.3–83.9%)	83.9% (71.7–92.4%)	78.5% (69.9–85.5%)
	LDBio neg	16	47			
	LDBio pos	44	9			
ImmunoCAP Asp IgG		CPA	Non-CPA	86.7% (75.4–94.1%)	80.4% (67.6–89.8%)	83.6% (75.6–89.8%)
	IgG neg	8	45			
	IgG pos	52	11			
Symptoms > 3 months with past history of PTB						
LDBio LFA		CPA	Non-CPA	74.1% (60.3–85%)	85% (70.2–94.3%)	78.7% (69.1–86.5%)
	LDBio neg	14	34			
	LDBio pos	40	6			
ImmunoCAP Asp IgG		CPA	Non-CPA	85.2% (72.9–93.4%	82.5% (67.2–92.7%)	84% (75.1–90.8%)
	IgG neg	8	33			
	IgG pos	46	7			
Symptoms > 4 weeks with lung cavity						
LDBio LFA		CPA	Non-CPA	68.9% (55.7–80.1%)	81.82% (59.7–94.8%)	72.3% (61.4–81.6%)
	LDBio neg	19	18			
	LDBio pos	42	4			
ImmunoCAP Asp IgG		CPA	Non-CPA	82% (70–90.6%)	86.4% (65.1–97.1%)	83.1% (73.3–90.5%)
	IgG neg	11	19			
	IgG pos	50	3			
Symptoms > 4 weeks excluding abpa						
LDBio LFA		CPA	Non-CPA	67.12% (55.1–77.7%)	89.77% (81.5–95.2%)	79.5% (72.4–85.5%)
	LDBio neg	24	79			
	LDBio pos	49	9			
ImmunoCAP Asp IgG		CPA	Non-CPA	82.19% (71.5–90.2%)	87.50% (78.7–93.4%)	85.1% (78.6–90.2%)
	IgG neg	13	77			
	IgG pos	60	11			
Symptoms > 4 weeks with bronchiectasis and excluding abpa						
LDBio LFA		CPA	Non-CPA	80% (65.4–90.4%)	86.67% (69.28–96.2%)	82.7% (72.2–90.4%)
	LDBio neg	9	26			
	LDBio pos	36	4			
ImmunoCAP Asp IgG		CPA	Non-CPA	86.7% (73.2–95%)	86.7% (69.3–96.2%)	86.7% (76.8–93.4%)
	IgG neg	6	26			
	IgG pos	39	4			

**Table 5 jof-08-00400-t005:** Sensitivity & specificity of LDBio LFA kit & ImmunoCAP Asp IgG (cut off >40 mgA/L).

Population/Test	No of Observations			Sensitivity	Specificity	Diagnostic Accuracy
Symptoms > 4 weeks						
LDBio LFA		CPA	Non-CPA	68.3% (55.04–79.74%)	75.4% (66.49–83.02%)	72.99% (65.75–79.43%)
	LDBio neg	19	86			
	LDBio pos	41	28			
ImmunoCAP Asp IgG		CPA	Non-CPA	75% (62.14–85.28%)	90.35% (83.39–95.08%)	85.06% (78.88–90.00%)
	IgG neg	15	103			
	IgG pos	45	11			
Symptoms > 4 weeks and prior history of PTB						
LDBio LFA		CPA	Non-CPA	75.51% (61.13–86.66%)	76.12% (64.14–85.69%)	75.86% (67.04–83.32%)
	LDBio neg	12	51			
	LDBio pos	37	16			
ImmunoCAP Asp IgG		CPA	Non-CPA	79.59% (65.66–89.76%)	91.04% (81.52–96.64%)	86.21% (78.57–91.91%)
	IgG neg	10	61			
	IgG pos	39	6			
Symptoms > 3 months with past history of tuberculosis						
LDBio LFA		CPA	Non-CPA	75.00% (59.66–86.81%)	74.00% (59.66–85.37%)	74.47% (64.43–82.91%)
	LDBio neg	11	37			
	LDBio pos	33	13			
ImmunoCAP Asp IgG		CPA	Non-CPA	77.27% (62.16–88.53%)	94.00% (83.45–98.75%)	86.17% (77.51–92.43%)
	IgG neg	10	47			
	IgG pos	34	3			
Symptoms > 4 weeks with lung cavity						
LDBio LFA		CPA	Non-CPA	69.23% (54.90–81.28%)	67.74% (48.63–83.32%)	68.67% (57.56–78.41%)
	LDBio neg	16	21			
	LDBio pos	36	10			
ImmunoCAP Asp IgG		CPA	Non-CPA	76.92% (63.16–87.47%)	96.77% (83.30–99.92%)	84.34% (74.71–91.39%)
	IgG neg	12	30			
	IgG pos	40	1			
Symptoms > 4 weeks excluding abpa						
LDBio LFA		CPA	Non-CPA	67.80% (54.36–79.38%)	82.35% (73.55–89.19%)	77.02% (69.74–83.27%)
	LDBio neg	19	84			
	LDBio pos	40	18			
ImmunoCAP Asp IgG		CPA	Non-CPA	76.27% (63.41–86.38%)	95.10% (88.93–98.39%)	88.20% (82.19–92.74%)
	IgG neg	14	97			
	IgG pos	45	5			
Symptoms > 4 weeks with bronchiectasis and excluding abpa						
LDBio LFA		CPA	Non-CPA	80.56% (63.98–91.81%)	71.79% (55.13–85.00%)	76.00% (64.75–85.11%)
	LDBio neg	7	28			
	LDBio pos	29	11			
ImmunoCAP Asp IgG		CPA	Non-CPA	80.56% (63.98–91.81%)	94.87% (82.68–99.37%)	88.00% (78.44–94.36%)
	IgG neg	7	37			
	IgG pos	29	2			

**Table 6 jof-08-00400-t006:** Characteristics of different studies reporting diagnostic performances of LDBio LFA.

Author	Country/Year	Study Type	Population	Comparator	Sensitivity	Specificity	Others
Piarroux et al. [15]	France/2019	Both retrospective and prospective	All samples received for *Aspergillus* serology (ABPA, CPA, IA SAIA *). Retrospective: 262 cases (68 CPA) & 188 controls. Prospective: 44 cases (11 CPA) & 213 non-cases	Who did not correspond to case definition of CPA	88.9%	96.3%	Definition of CPA as per ERS/ESCMID
Hunter et al. [8]	UK/2019	Cross sectional	CPA patient sera. 154 CPA patients, 150 healthy controls.	Healthy control	91.6%	98%	Definition of CPA as per ERS/ESCMID
Rozaliyani et al. [19]	Indonesia/2020	Prospective	Adults with symptoms after completing tuberculosis therapy.	Patients without diagnosis CPA	80%	70%	Sputum for fungal culture was used as an essential diagnostic criterion.
Ray et al. (present study)	India/2021–22 present study	Prospective	Patients presenting with respiratory symptoms > 4 weeks. 74 CPA & 100 non-CPA patients	Patients being tested who did not have CPA	67.6%	81%	Definition of CPA as per ERS/ESCMID

* Invasive or sub-acute invasive aspergillosis.

## Data Availability

The dataset is available with the authors.

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
