# Peer review of "Efficacy of LD Bio Aspergillus ICT Lateral Flow Assay for Serodiagnosis of Chronic Pulmonary Aspergillosis"

_jof, 2022, doi:10.3390/jof8040400_

Round 1
Reviewer 1 Report
Nice study. Please work on small mistakes, abbreviations, and presentation.
Minor comments:
Lines 158-166: CI seems incorrect "81% (95% CI: 71.9%-81%)", perhaps a typo for 91%? Same in table 4 and abstract. Please double check your data. Also, stating that sens and spec 'improved' while CI overlap, does not seem accurate. Please leave out the word improved in result section (I do not mind the use in discussion).
table 1; lots of abbreviations not explained in table subtext. Layout messed up which makes some content unreadable. This is the case for many tables, perhaps due to pdf converter?
table 3; Serum galactomannan vs BAL GM in non-CPA group is discrepant, serum GM very high. Also, both groups very low positive fungal culture compared to BAL/serum GM. Can you discuss these numbers? Please add invasive fungal diagnoses.
Change -ve and +ve into negative and positive throughout the paper (or pos/neg and list in table subtext).
Conflict of interest is full of typos.
Doi's are included but ref 6 does not show author/title: 2021.10.04.21264524v1.full.pdf. Accessed February 15, 2022. https://www.medrxiv.org/content/10.1101/2021.10.04.21264524v1.full.pdf
Author Response
Reviewer 1
Comment 1:
Lines 158-166: CI seems incorrect "81% (95% CI: 71.9%-81%)", perhaps a typo for 91%? Same in table 4 and abstract. Please double check your data. Also, stating that sens and spec 'improved' while CI overlap, does not seem accurate. Please leave out the word improved in result section (I do not mind the use in discussion).
Response 1:
Yes, we have double checked the data & the marked CI was a clerical mistake, it has been changed to 88.2% in the respective areas. The word 'improved' has been removed as suggested from the result section
Comment 2:
table 1; lots of abbreviations not explained in table subtext. Layout messed up which makes some content unreadable. This is the case for many tables, perhaps due to pdf converter?
Response 2:
Yes , the abbreviations have been added to the tables & the layout has been corrected.
Comment 3:
table 3; Serum galactomannan vs BAL GM in non-CPA group is discrepant, serum GM very high. Also, both groups very low positive fungal culture compared to BAL/serum GM. Can you discuss these numbers? Please add invasive fungal diagnoses.
Response 3:
All together 10 & 7 patients had elevated BAL & Serum Galactomannan respectively in the Non-CPA group. This may be due to false positive reactions brought about by antibiotics or food habits. This has been mentioned in the discussion section as "A significant proportion of controls in our study was also positive for serum galactomannan indicating a plausibility of false positivity owing to previous antibiotics or food habits.23-24"
This finding needs to be further corroborated in future studies from multiple centers in India.
Comment 4:
Change -ve and +ve into negative and positive throughout the paper (or pos/neg and list in table subtext).
Response 4:
+ve/-ve have been changed to pos/neg in the table layout
Comment 5:
Conflict of interest is full of typos.
Response 5:
Typing errors in the conflict of interest section have been corrected.
Comment 6:
Doi's are included but ref 6 does not show author/title: 2021.10.04.21264524v1.full.pdf. Accessed February 15, 2022. https://www.medrxiv.org/content/10.1101/2021.10.04.21264524v1.full.pdf
Response 6:
The mentioned reference have been corrected as per recommendations
Reviewer 2 Report
This study assessed the efficacy of LD Bio Aspergillus ICT Lateral flow assay (LDBio LFA) for the diagnosis of chronic pulmonary aspergillosis (CPA) in India – a place with high prevalence of tuberculosis. They found that the estimated sensitivity and specificity of LDBio LFA was 67.6% (95% CI: 55.7%- 78%) and 81% (95% CI: 71.9%-81%) respectively which increased to 73.3% (95% CI: 60.3%-83.9%) and 83.9% (95% CI: 71.7%-92.4%) respectively in patients with past history of tuberculosis. They suggested that LDBio LFA is a point-of-care test with reasonable sensitivity and specificity. Overall, the manuscript is well-written. I just have several comments.
- Please add more description about LDBio LFA, such as turn around time.
- Please briefly describe the clinical and radiologic diagnostic criteria of CPA in this study.
- Many abbreviations were used in the tables. Please add the note to explain.
- Please specifically describe “MDI use” in table 1. Do you want to mean inhaled corticosteroid?
- Please added the medication, such as systemic corticosteroid and immunosuppressant in the table 1.
- Please organized the all tables again. The present form is difficult to follow.
Author Response
Reviewer 2
Comment 1:
Please add more description about LDBio LFA, such as turn around time.
Response 1:
Details regarding the the storage , working principle & interpretation has been added to the methods section as following "The Aspergillus ICT IgG IgM lateral flow assay (LDBio LFA, Diagnostics, Lyon, France) in a cassette format was used to test each sample, and all tests were performed and interpreted according to the manufacturer's instructions. The required number of cassettes were removed from storage at 4°C, brought to room temperature and labelled. 15 µl of serum was added to the sample well of each cassette using a calibrated micropipette and sterile disposable tips, followed by 4 drops of eluent dispensed directly from the dropper. The cassettes were then allowed to stand for 20 minutes, and the test results were read between 20 and 30 minutes after adding the eluent to the last cassette. Accounting for pre-test centrifugation of blood samples to separate serum and proper pre- and post-test documentation, the turnaround time of the test (from receiving a blood sample in the laboratory to availability of the report online) is therefore between 30 and 45 minutes"
Comment 2:
Please briefly describe the clinical and radiologic diagnostic criteria of CPA in this study.
Response 2:
Clinical & radiological criteria for diagnosis has been updated as per ESCMID criteria, as follows ". Clinical parameters includes the presence of at least one of these symptoms, such as hemoptysis, cough, exhaustion, chest discomfort, and/or dyspnea, for more than three months. Radiographic findings consistent with CPA (at least one of cavitation and/or fungal ball confirmed by CT scan) should be present ".
Comment 3:
Many abbreviations were used in the tables. Please add the note to explain.
Response 3:
Abbreviations has been added as notes to the tables.
Comment 4:
Please specifically describe “MDI use” in table 1. Do you want to mean inhaled corticosteroid?
Response 4:
MDI use have been added as including short or long acting bronchodilators & inhaled corticosteroids in the abbreviation notes added to the table(table 1)
Comment 5:
Please added the medication, such as systemic corticosteroid and immunosuppressant in the table 1.
Response 5: Though 7 patients (ILD:1 Sarcoidosis:5 Chronic HSP:1) were on immunosuppressants, our proforma did not specifically capture the exact drugs consumed. So, we do not have the said information.
Comment 6:
Please organized the all tables again. The present form is difficult to follow
Response 6:
The tables have been organized, might have had been disorganized due to pdf converter used later.
Round 2
Reviewer 2 Report
The authors response well, so I have no mroe comment.